# Genome Editing for Cystic Fibrosis

**DOI:** 10.3390/cells12121555

**Published:** 2023-06-06

**Authors:** Guoshun Wang

**Affiliations:** Department of Microbiology, Immunology and Parasitology, Louisiana State University Health Sciences Center, CSRB 607, 533 Bolivar Street, New Orleans, LA 70112, USA; gwang@lsuhsc.edu; Tel.: +1-(504)-568-7908; Fax: +1-(504)-568-8500

**Keywords:** cystic fibrosis, CRISPR/Cas, genome editing, gene editing, CFTR

## Abstract

Cystic fibrosis (CF) is a monogenic recessive genetic disorder caused by mutations in the CF Transmembrane-conductance Regulator gene (*CFTR*). Remarkable progress in basic research has led to the discovery of highly effective CFTR modulators. Now ~90% of CF patients are treatable. However, these modulator therapies are not curative and do not cover the full spectrum of *CFTR* mutations. Thus, there is a continued need to develop a complete and durable therapy that can treat all CF patients once and for all. As CF is a genetic disease, the ultimate therapy would be in-situ repair of the genetic lesions in the genome. Within the past few years, new technologies, such as CRISPR/Cas gene editing, have emerged as an appealing platform to revise the genome, ushering in a new era of genetic therapy. This review provided an update on this rapidly evolving field and the status of adapting the technology for CF therapy.

## 1. Introduction

CRISPR—clustered regularly interspaced short palindromic repeats—were first discovered in the 3′-end flanking region of the *iap* gene in *E. coli* in 1987 [1] and later found in many other bacteria and even archaea [2,3,4]. These repeats and their associated proteins constitute an adaptive immune system to protect host microbes from invasion by foreign genetic elements, such as viruses, through targeted cleavage of the nucleic acids with RNA-guided endonuclease [5,6,7,8]. Among many types of CRISPR-associated proteins (Cas), *Streptococcus pyogenes* Cas9 (SpCas9) is the most studied [9,10]. Jennifer Doudna, Emmanuelle Charpentier, and their teams first engineered the CRISPR/Cas9 system for programmable gene editing in 2012 [11]. Then in the following year, Feng Zhang, George Church, and their colleagues published the first eukaryotic genome engineering with the system [12,13], which opened the floodgates for genome engineering in living cells of diverse species. This breakthrough has led to revolutionary advances in medicine through the direct correction of disease-associated genes in the genome for therapy. To date, a decade after the invention of programmable gene editing, there have been 76 clinical trials registered with the US Food and Drug Administration (ClinicalTrials.gov) to test CRISPR/Cas technology in different diseases or conditions. Excitingly, the first such trial for sickle cell disease and β-thalassemia achieved a sustained cure for these two deadly transfusion-dependent blood diseases [14], greatly inspiring basic researchers and clinical practitioners to pursue similar treatments for other genetic diseases, including cystic fibrosis (CF).

## 2. CF Gene Mutations and CF Clinical Diseases

### 2.1. CF Genomic Mutations

CF is one of the most common life-threatening genetic disorders, affecting all races and ethnic groups, but the Caucasian population has the highest rate of incidence [15]. There are currently over 100,000 CF patients worldwide [16]. CF genetic lesions have been mapped to human chromosome 7 (7q31.2) in 1989 by Francis Collins and Lap-chee Tsui’s groups [17,18]. The normal function of the CF locus is to direct the synthesis of CF transmembrane-conductance regulator (CFTR) protein, a cAMP-activated anion channel [19]. Since the discovery of *CFTR* [18,20], more than 2000 mutations have been identified, which cause clinical illnesses with varying severity [21,22]. Based on CFTR protein production and channel activity, CF mutations can be categorized into five classes [23,24]. Class 1 contains nonsense mutations, e.g., G542X, giving rise to premature termination of CFTR protein. Class 2, represented by F508del, produces a full-length, but misfolded CFTR protein that degrades in the endoplasmic reticulum. Class 3, such as G551D, results in a full-length CFTR with a null channel function. Class 4 leads to a mature, correctly targeted CFTR with partial channel function. Class 5 produces insufficient CFTR protein. The F508del mutation in one or two alleles occupies ~85.5% of all CF individuals, and the other CFTR mutations take up the remaining ~14.5% [25].

### 2.2. CF Clinical Diseases

Clinically, CF presents as a progressive, chronic, and debilitating illness [26] that affects multiple organ systems, including the lung, sinuses, gastrointestinal tract, liver, pancreas, and vas deferens. Among all these afflicted organ systems, CF lung disease is predominant in the adult patient population, accounting for over 90% of CF morbidity and mortality. CFTR transports chloride and bicarbonate anions across the apical membrane of airway epithelia and regulates sodium absorption and airway surface liquid hydration [27]. CFTR dysfunction impairs anion secretion, increases sodium absorption, causes airway surface liquid dehydration, and disables mucociliary clearance [28]. It is widely believed that these conditions induce chronic bacterial infection, persistent neutrophilic inflammation, and small airway mucopurulent obstruction, which ultimately cause bronchiectasis and pulmonary failure.

The pancreas is one of the earliest- and most commonly-affected organs in people with CF. The term “cystic fibrosis” was named after the fibrocystic lesions observed in the pancreas from pediatric autopsy cases [29]. Ductal obstruction leads to pancreatic protein buildup behind the obstructive sites, which induces inflammation, fibrosis, fatty replacement, and eventually pancreatic destruction. Approximately 85% of CF patients become pancreatic insufficient, requiring lifelong pancreatic enzyme replacement [30]. CF-related diabetes is an increasingly recognized complication of CF that occurs in 40–50% of adults with CF [31].

CF intestinal disease begins in the uterus and is often the first sign of CF encountered clinically. Meconium ileus (MI) presents in up to 20% of neonates with CF [32]. The inspissated meconium obstructs the small intestine at the terminal ileum. If left untreated, the prognosis is poor. In infancy and childhood, CF patients demonstrate nutrient malabsorption, steatorrhea, and inadequate weight and body length, for which pancreatic insufficiency is largely responsible [33]. Pancreatic enzyme replacement therapy proves beneficial in overcoming some of these problems. However, distal intestinal obstruction syndrome (DIOS) and constipation continue to be a problem throughout the patients’ lives [34]. Adult CF patients demonstrate small intestine bacterial overgrowth (SIBO) [35] and large intestine dysbiosis [36,37]. CF intestinal disease is also marked by a swollen intestinal wall, intestinal stricture, excessive neutrophil infiltration, and fibrosing colonopathy [38,39,40,41,42,43,44].

CF liver disease (CFLD) affects ~30% of CF patients. Clinical manifestations of CFLD are elevation of serum liver enzymes, hepatic steatosis, focal biliary cirrhosis, multilobular biliary cirrhosis, neonatal cholestasis, cholelithiasis, cholecystitis, and micro-gallbladder [45]. CFTR chloride channel is expressed to the apical membrane of cholangiocytes lining the biliary ducts [46], essential to maintain pH regulation and biliary HCO_3_^−^ secretion. A proper alkaline balance is believed to protect cholangiocytes against hydrophobic bile acids [47].

CF damages the reproductive system, with a majority of adult male patients showing congenital bilateral absence of vas deferens [48]. Females with CF are found less fertile than normal healthy women, and some show congenital absence of the uterus and vagina [49].

The CFTR channel is also expressed in many non-epithelial cells, including endothelial cells [50], skeletal muscle [51], smooth muscle [52,53], cardiac muscle [54], red blood cells [55], platelets [56], neurons [57] and even reproductive sperm cells [58], suggesting a role of CFTR in the functions of these cells. More importantly, immune cells, such as neutrophils and monocytes/macrophages, have CFTR expression [59,60,61]. As infection and inflammation are the two major pathologies in CF, it was predicted that the host immune system is impaired. Indeed, CFTR is a major chloride channel that transports chloride to phagosomes to produce hypochlorous acid, a potent antimicrobial oxidant [61,62,63,64,65,66]. CFTR loss of function in neutrophils impairs the host’s defense against selective pathogens and compromises the host’s ability to resolve inflammation [59,67,68,69,70]. Other abnormalities are also reported in CF neutrophils, including suboptimal activation [71], cleavage of CXCR1 [72], hyper-sensitivity to LPS stimulation [73], deviant production of reactive oxygen species [74], genome-wide gene expression perturbation [75], alteration in inflammatory signaling [76], hyper-production of IL-8 [77,78], delayed apoptosis [79], abnormal extracellular trap formation [80], hyper-oxidation of glutathione [81], and lately abnormal granule release [82]. Moreover, CF monocytes/macrophages exhibit deficiencies in their host defense and other functions, including hyper-sensitivity to challenge [83,84,85], impaired capacity of killing internalized bacteria [60,86,87], and reduced scavenger ability [88]. In addition to the infection and inflammation complications, CF patients often suffer arthropathies [89,90]. CF-related arthropathy has a rate of occurrence ranging from 2% to 8.5% [91] with distinct symptoms, such as recurrent episodes of joint pain, swelling, tenderness, and limitation of movement. One or more joints may be affected, and there may also be fever and skin manifestations [90]. Moreover, 2–7% of CF patients suffer hypertrophic pulmonary osteoarthropathy with long-bone periostitis, and finger and toe clubbing with symmetric arthralgia [92,93]. These inflammation-related clinical manifestations further demonstrate the CF immune defect.

## 3. General Considerations in Gene Editing Design for CF

First, CF is a systemic disease that affects many epithelial and non-epithelial organs. What is the best route to deliver gene editing? From a therapeutic perspective, the ideal way is through systemic delivery that can reach all tissues with one application. Second, CF has more than 2000 distinct mutations. How can gene editing achieve pan-mutation correction with one formula? One strategy is to perform targeted insertion of a full-length *CFTR* cDNA into the CFTR locus to replace the endogenous *CFTR*. This one-size-fits-all approach can target almost all CF mutations. Third, terminally differentiated epithelial cells have a certain life span. How can durable gene editing be achieved? Stem and progenitor cells would be the preferential targets for correction. Fourth, CF clinical diseases develop early in life. What is the best age to perform gene editing for therapy? It is apparently important to target CF before damage to organ structures and functions become irreversible. Thus, gene editing components and formulation must be applicable to young patients. Fifth, safe genome manipulation is paramount. Efforts to minimize off-target effects and to ensure long-term safety should guide any gene editing design. Attention should be specially paid to the systemic delivery of gene editing to young patients, as germline modification may occur.

## 4. Gene Editing Tools for Selection

### 4.1. Cas Nuclease Editors

There are 2 classes of CRISPR/Cas systems, which can be subdivided into 6 types and at least 29 subtypes according to their structure and function of Cas protein [94]. Class I contains Types I, III and IV, while Class 2 contains Types II, V and VI. Each member in Class I has multiple effector complexes with several Cas proteins, whereas each Class II member has a single, large, multidomain Cas protein. Due to the relatively simple ribonucleoprotein complexes involved, Class II members have been extensively studied and are the most widely used tool for genome engineering.

The prototype CRISPR/Cas9 complex is an assembly of CRISPR RNAs and Cas9 endonuclease. The CRISPR RNAs have two separate molecules: (1) guiding CRISPR RNA (crRNA) [95], and (2) trans-acting CRISPR RNA (tracrRNA) [96]. The crRNA recognizes its target DNA sequence and tracrRNA recruits Cas9 to the target sites. Cas9 induces double-strand breaks (DSBs) through its two distinct nuclease domains: an HNH-like nuclease domain that cleaves the gRNA-targeted strand, and a RuvC-like nuclease domain to cut the non-target strand [97]. To simplify the system, the two separate RNAs can be fused into a single RNA chimera, referred to as single-guide RNA (sgRNA), which is able to achieve a similar RNA-guided DNA targeting function [11,98]. SpCas9, a 1368 amino-acid protein from the Class II, Type II category, induces a blunt DSB at the target sequence positioned next to 5′-NGG (N represents any nucleotide) protospacer adjacent motif (PAM) [10]. To increase the availability of target sites and the target specificity, multiple SpCas9 variants have been generated [99], one of which is xCas9 which recognizes 5′-NG, 5′-GAA, and 5′-GAT PAM sequences [100]. To modify the Cas9 enzymatic property, a single point mutation, either D10A or H840A, was introduced into Cas9 to turn it into a nickase (nCas9) that can only cleave single-stranded target sites recognized by gRNA. In contrast to the regular SpCas9 that uses a single sgRNA to produce a DSB, nCas9 requires two sgRNAs to achieve a DSB, thus enhancing the targeting specificity. Another Cas9 derivative is “dead” Cas9 (dCas9) which has a D10A and H840A double mutation. As a result, both HNH- and RuvC-nuclease domains are disabled. This dCas9 binds to its gRNA target site as normal, but not induces breakage in either DNA strand. Thus, the dCas9 can be fused to transcriptional activators or suppressors to regulate gene transcription [101,102].

In addition to SpCas9, a battery of Cas9 orthologues from other bacterial species have been discovered. These orthologues have different PAM sequences that provide a greater choice of target sites across the human genome [103]. Among them are the Cas9 from *Streptococcus thermophiles*, *Neisseria meningitidis*, *Staphylococcus aureus* and *Campylobacter jejuni*. These different variants of Cas9 offer expanded genome targeting capabilities, improved specificity, and biochemical properties [103,104].

Cas12 and Cas13 from Type V and Type VI categories, respectively, are two latest additions to the gene editing toolbox. Cas12a nucleases from *Acidaminococcus* spp. (AsCas12a) and *Lachnospiraceae* spp. (LbCas12a) recognize DNA target sequences with complementarity to the crRNA spacer positioned next to a 3′ PAM [105], and generate a staggered DNA double-strand break by a RuvC domain and a putative nuclease (Nuc) domain. This unique feature makes it ideal for multiplexed genome editing [106]. Cas13 nucleases solely target single-stranded RNA without altering the DNA, which can be selected for transcriptomic manipulation [107].

### 4.2. Base Editors

Base editing uses a different design to achieve genome modification by the direct generation of precise point mutations. DNA base editors comprise a catalytically impaired Cas nuclease fused with a base modification enzyme that induces base alteration on a single strand of DNA. Then the cellular DNA repair machinery intervenes to repair the mismatch on the complementary strand to complete the intended base conversion [108]. There are two classes of DNA base editors that have been reported: (1) cytosine base editors that convert a C•G base pair into a T•A base pair, and (2) adenine base editors that convert an A•T base pair into a G•C base pair [109].

### 4.3. Prime Editors

Prime editing enables all types of DNA substitutions, small insertions, and small deletions to be created at targeted sites in the genome [110]. It differs from the CRISPR/Cas9 editing and base editing systems in the following aspects: (1) prime editing does not require DSBs; (2) the prime editor protein is a fusion of nCas9 and a reverse transcriptase; and (3) prime editing guide RNA (pegRNA) contains a specific genome-targeting sequence and a programmable reverse transcriptase template to introduce the desired edit [111]. Compared to Cas nucleases and base editors, prime editors offer several advantages: (1) High precision. As the desired edit is programmed within the pegRNA template, and the targeted DNA is only nicked on one strand, the achieved base change is very specific and rarely generates insertion or deletion mutations. (2) Flexibility. Prime editing can change bases relatively far from the PAM site and is thus less restricted by PAM site availability [112]. (3) Lower cellular replication dependence. Prime editors do not rely on homologous recombination machinery to introduce the desired edit and are therefore effective in any phase of the cell cycle [111].

## 5. CRISPR-Based CFTR Gene Editing

### 5.1. In Vitro Correction of CFTR Mutations

Schwank and colleagues first reported using the CRISPR/Cas9 editing system to correct the CFTR locus in cultured intestinal stem cells of F508del-homozygous CF patients [113]. The gene-corrected stem cells were able to develop into organoids that functionally responded to forskolin by volume swelling. Firth et al. derived induced pluripotent stem cells (iPSCs) from skin fibroblast cells from homozygous F508del CF patients, and performed CRISPR/Cas9 gene correction in *CFTR*. The corrected iPSCs were able to differentiate into mature airway epithelial cells and showed restoration of CFTR-specific chloride transport function [114]. In addition to the prevalent F508del mutation, premature stop codon mutations, such as c.1679 + 1634A > G (1811 + 1.6 kbA > G) and c.3718-2477C > T (3849 + 10 kbC > T), or c.3140-26A > G (3272-26A > G), were corrected by CRISPR/Cas9 editing [115]. Furthermore, the targeted insertion of a full-length *CFTR* cDNA into the CF locus was accomplished in airway stem cells via CRISPR/Cas9 editing [116]. Similarly, Hu and his team have successfully integrated a full-length *CFTR* cDNA into GGTA1, a safe harbor gene in cultured pig cells [117]. These data demonstrate the possibility of pan-mutation correction with one gene editing vector.

Using adenine base editors, Krishnamurthy et al. successfully corrected premature stop codon mutations [118]. Moreover, prime editors were recently tested in the functional correction of CF organoids [119]. These seminal experiments clearly demonstrate that different gene editing tools are capable of correcting the basic CF defect in vitro.

### 5.2. Creation of CF Models

When discussing gene editing for CF, it is hard not to mention the many CF models created using this technology. Because of the simplicity in design and high efficiency in genome manipulation, the gene editing approach has become mainstream in the production of new CF models from cell lines to live animals.

Multiple CF cell lines have been generated using the Cas9 nucleases, including Calu-3 CF cells [120], F508del-CF HL-60 cells [121], F508del-CF T84 cells [122], 16HBE14o- CF cells with F508del, G542X or W1282X mutations [123], CFTR-/- IPEC-J2 porcine cell line [117], and G551D-CF Caco-2 cells [124]. As compared to existing CF cell lines that were immortalized from patients’ tissues, the gene-edited cell lines have their own isogenic controls, thus providing a paired cell culture system for CF basic research and drug selection.

Based on the superior genome-targeting and gene-editing efficiency, direct pronuclear injection or nucleofection of gene editors into fertilized eggs at the single-cell stage (zygote), followed by after-birth screening, is established as a protocol to produce animal models. Using the strategy, multiple CF animal models have been generated, including G542X mice and rats [125,126,127], F508del rabbits [128], and G551D ferrets [129]. Furthermore, by combining CRISPR/Cas9 gene editing and somatic cell nuclear transfer technologies, F508del and G542X CF lambs have been produced. These animal models add valuable selections to the existing CF animal pool to facilitate the study of CF pathogenesis and the development of novel therapies.

### 5.3. In Vivo Correction of CFTR Mutations

The concept of using different gene editing tools to convey genomic revision of *CFTR* mutations has been proven by numerous in vitro gene correction cases, as discussed above. Now, the compelling call is to adapt the technology for in vivo CFTR correction as a therapy. As CF lung disease is the major complication in CF patients, early explorations of in vivo gene editing related to CF have been focused on the lung. McCray and his colleagues tested a group of amphiphilic shuttle peptides to facilitate gene editor delivery to mouse lungs and achieved editing of loxP sites in airway epithelia of ROSA^mT/mG^ mice [130]. The editing efficiency was ~13% in the large airways and ~12% in the small airways. The same team also delivered a base editor to Rhesus monkey lungs [131], and achieved editing efficiencies of up to 5.3% in rhesus airway epithelia. In mice, a similar procedure led to the persistence of the gene-edited epithelia for up to 12 months. These exciting early data demonstrate the therapeutic potential and feasibility of using CRISPR-based gene editors for CF lung therapy. It is anticipated that the next step would be applying therapeutic editors to CF animal models to determine if CF lung phenotypes can be rescued.

## 6. Non-CRISPR-Based CFTR Gene Editing

Although this review is tasked with updating the advances in CRISPR-based gene editing for CF, new developments on the non-CRISPR gene-editing front are worth discussion and comparison. Early non-CRISPR editors include mainly zinc finger nuclease- (ZFN-) based and transcription activator-like effector nuclease- (TALEN-) based. Although currently fewer labs are using these systems for CF gene editing, progress is still being made. Suzuki et al. electroporated ZFN mRNA into CF patient-derived airway basal cells, followed by AAV-6 donor transduction, and achieved efficient functional correction of CFTR [132]. Xia et al. packaged the TALEN system into helper-dependent adenoviral vectors and ~5% targeted gene integration was obtained [133]. In addition to these conventional non-CRISPR editor systems, Egan and her team have adapted a novel peptide nucleic acid (PNA) gene editing system for CFTR gene correction [134,135]. The PNA system was first reported by Rogers et al. to achieve site-specific DNA recognition and to mediate site-directed recombination [136]. Different from the CRISPR-nuclease-triggered endogenous DNA repair, PNAs form a triplex structure with DNA by strand invasion and prompt DNA repair and recombination of short donor DNA [137]. When PNAs and donor DNAs are packed into biodegradable polymer nanoparticles, F508del-CF airway epithelial cells can be targeted and corrected [134]. Notably, using this approach Piotrowski-Daspit and colleagues have achieved in vivo correction of F508del-CFTR mutation in the CF mouse model through systemic delivery [138]. Gene editing has been found in multiple epithelial tissues, including the nasal epithelium, trachea, lung, ileum, colon, and rectum, with a correction rate varying from ~0.1% to ~2%. Impressively, phenotypes such as CF lung inflammation and epithelial chloride transport are shown to be improved.

## 7. Comparison of CRISPR-Editors and Non-CRISPR PNA Editors for CF Gene Editing

Previous publications have compared the CRISPR editing system with the non-CRISPR TALEN or ZFN system for CF gene correction [135,139]. Here we focus our comparison of the CRISPR system with the new non-CRISPR PNA one. Both systems have been proven feasible in CFTR gene correction in vitro and in vivo, indicating the possibility of rescuing CF defects. However, they are still at their early development stage for in vivo applications, and further optimization and better understanding are clearly needed. First, the rates of editing or correction are still low for a complete rescue of CF. Second, the spectrum of target cells needs a better understanding. Third, the durability of gene editing requires longer observation and inspection. Fourth, long-term safety has not been investigated. Nevertheless, these two approaches are re-igniting the hope of a better CF genetic therapy. Because of their distinct acting mechanisms in genome modification, each approach has its pros and cons, which might be important to consider for future preclinical and clinical trials. First, the CRISPR-based approach needs suitable PAM sites, but the non-CRISPR-based approach does not. Thus, the latter approach may provide better flexibility. Second, the CRISPR-based approach does not have to rely on endogenous DNA repair machinery, but the non-CRISPR-based approach requires this machinery. For example, the base-editing or prime-editing complex carries its own DNA modification enzyme, which may offer a higher editing efficiency. Third, the non-CRISPR editor is small and can fit into nanoparticle carriers. Thus, it can readily pass through tissue barriers to target the epithelial layer. This merit makes systemic delivery possible. Fourth, the non-CRISPR editor does not have high immunogenicity, thus making repetitive administration possible.

## 8. Barriers to Overcome to Improve In Vivo Gene Editing Efficiency for CF

From the existing data, there is an apparent gap in gene editing efficiency between in vitro and in vivo applications in both CRISPR- and non-CRIPSR-based gene editors. Thus, delivery is a determining factor in the success of translating this new technology into any clinical benefits. As CF predominantly affects epithelium-lined organs, directing CF gene editors to the epithelial cells is believed to be essential for potential CF therapy. There are only two feasible routes to deliver the gene editors: (1) through epithelial lumens, and (2) through the circulation. Each route has its own barriers to overcome.

### 8.1. Gene Editor Delivery through Epithelial Lumens

For a luminally released gene editor to reach the nucleus of a target epithelial cell, multiple layers of barriers need to be overcome (Figure 1). First, on the top of an epithelium, there usually exists a mucus and/or surface liquid layer that gene editors have to penetrate. Second, an epithelium-lined organ usually evolves a mucociliary clearance mechanism for the purpose of host defense. This mechanism can also act to clear luminally delivered gene editors. Third, epithelial cells are typically polarized, and their subcellular structures, e.g., cytoskeleton, in the apical and basolateral compartments differ, which limits gene editor entry from the apical side. Fourth, epithelial stem and progenitor cells need to be targeted for any lasting gene editing. These cells sometimes are anatomically located underneath other epithelial cells and are hard to reach without losing the overlaying cells.

### 8.2. Gene Editor Delivery through the Circulatio

Circulation delivery has the potential of targeting every type of cell across the entire body. However, several barriers have to be overcome to achieve any high-level gene editing (Figure 2). First, intravenously released gene editors have to permeate the endothelial layer and escape the circulation to enter the interstitial space. Second, the escaped gene editors need to diffuse through the interstitial space and meshwork to reach the epithelial basement membrane. Third, penetration of the epithelial basement membrane will allow the gene editors to target the epithelial cells from the basolateral side.

Altogether, different routes of delivery dictate the required features in gene editor and vehicle designs. Generally speaking, a smaller size of a gene editor complex favors its penetration and diffusion to reach the target cells. Epithelial and endothelial junctions can be temporally manipulated to facilitate the escape of the editor complex. Vehicles can be engineered to enhance their penetrating and targeting capacity, such as the surface expression of shuttle peptides or receptors/ligands.

## 9. Prospects for a Cure in CF

Genetic therapy for CF has been pursued for decades [135,139]. The early strategy was focused on gene addition by supplementing the diseased cells with a functional copy of *CFTR* [140,141,142]. Multiple vectors have been developed and tested pre-clinically or clinically in the lungs, including adenoviral vectors [143,144], adeno-associated viral vectors [145,146,147,148], lentiviral vectors [149,150,151,152], and non-viral vectors [153]. Unfortunately, clinical improvement of lung functions has never been achieved. Retrospective reflection often points to multiple issues, including vector transduction efficiency, stem and progenitor cell targetability, and duration of the transgene expression [141,154]. However, a key and largely overlooked issue is that we do not know whether restoration of CFTR function in epithelial cells alone is sufficient to rescue CF clinical diseases. Nevertheless, there are several outstanding merits of this strategy. First, gene addition by providing a full-length CFTR cDNA can treat all CF mutations at once. Second, after decades of improvements in vector design and delivery procedure, impactful restoration of chloride transport defect across CF epithelia in vivo can be achieved. Third, persistent gene transfer is doable with integrating vectors. These merits maintain gene addition as a viable platform for CF therapy. Moreover, the developed vector systems can be repurposed to deliver gene editors into target cells and organs [155].

The emerging CRISPR-based gene editing technology and the new non-CRISPR PNA-based gene editing technology offer precise and versatile options for genome manipulation, which enable the correction of CF mutations in their natural locations on the chromosome. After this correction, CFTR gene function and regulation should be restored to normal levels. These strategies hold the great promise of achieving an ultimate cure for CF. With the principle proven, translation of the technology into a CF therapy boils down to the following issues: (1) what vehicle to use to best deliver the gene editors, (2) how to effectively target the widespread diseased organ systems, (3) when to perform gene editing to achieve maximal clinical benefits, and (4) whether gene editing is durable and safe. Addressing these issues will bring a true cure for CF within reach.

## Figures and Tables

**Figure 1 cells-12-01555-f001:**
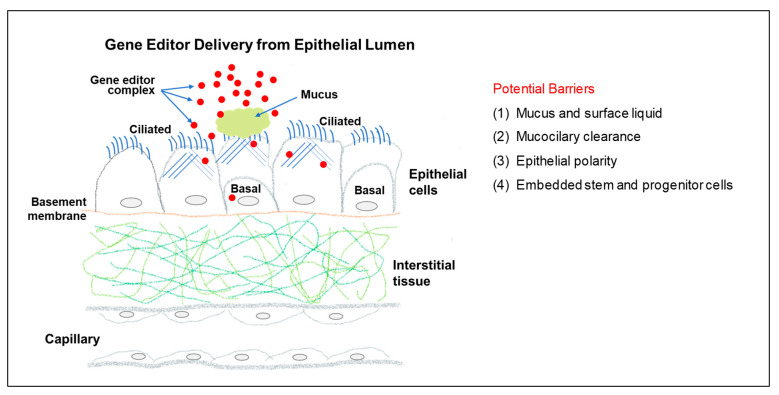
Gene editor delivery from an epithelial lumen. Potential barriers to overcome to achieve highly efficient gene editor delivery from an epithelial lumen.

**Figure 2 cells-12-01555-f002:**
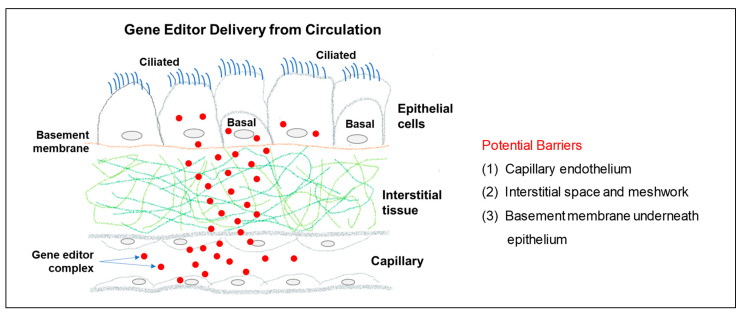
Gene editor delivery via circulation. Potential barriers to overcome to achieve highly efficient gene editor delivery from the circulation.

## Data Availability

Not applicable.

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
