# Peer review of "Genome Editing for Cystic Fibrosis"

_cells, 2023, doi:10.3390/cells12121555_

Round 1
Reviewer 1 Report
The review article “CRISPR-Based Genome Editing for Cystic Fibrosis” by Dr. Guoshun Wang provides a brief review on the mutations of CFTR, CF clinical diseases and gene-editing endeavors to correct mutated CFTR genes. In the first part, the author reviewed the different classes of CFTR mutations and different CF phenotypes. This part is redundant because there are tons of review articles out there already. The 2nd part focus on different gene-editing tools, which is not hard to find in the literature either. The last part described the current efforts on the correction of CFTR genomic mutations. However, the latter description is so brief without any in depth analysis of the pros and cons of the current methods used in CF genome editing. What needs to be improved if different strategy to be employed to treat CF disease.
Author Response
Thank you for your evaluation of our manuscript and providing constructive comments on revision.
Comment: the latter description is so brief without any in depth analysis of the pros and cons of the current methods used in CF genome editing. What needs to be improved if different strategy to be employed to treat CF disease.
Response: This review is tasked with updating the field with the recent advances in gene editing. Thus the author did not expand on the section that had been reviewed by others. However, to compare with different strategies, a new section on non-CRISPR-based CFTR gene editing has been included (Lines 277 - 322).
Reviewer 2 Report
Wang et. al. introduced the background of cystic fibrosis diseases and CRISPR/Cas gene editing tool. The research and clinical need for CF is summarized, indicating the great potential of CRISPR technolegies. The information is interesting and highly applicable. However, the scope and novelty of this review manuscript should be highlighted clearly. The introduction of other gene editing tools for CF should also be included. Why CF research will benifit from CRISPR/Cas? The technical challenges to overcome is not clear. There is not much detail of the CRISPR application from literature. A review manuscript should be organized well, and of great significance for the development of a topical area. For these reasons, this manuscript requires systematical revision before publishing.
Minor editing of English language required
Author Response
Thank you for reviewing our manuscript and providing the critiques.
Comment: The introduction of other gene editing tools for CF should also be included. Why CF research will benifit from CRISPR/Cas? The technical challenges to overcome is not clear. There is not much detail of the CRISPR application from literature. A review manuscript should be organized well, and of great significance for the development of a topical area. For these reasons, this manuscript requires systematical revision before publishing.
Response: The author apologizes for not stating it clearly in the original communication. This invited review is a part of a series on CRISPR-based gene editing. We are tasked to focus on CF gene editing. It is not our intention to make it a comprehensive review, but an update of the latest development in the field.
Per your suggestion, we have included a section on other gene editing tools and to compare the pros and cons of the different approaches (Lines 277-322, and Lines 324-340).
Significance and benefits of CRISPR/Cas in CF research has been put upfront in the abstract (Lines 15-18).
Technical challenges to overcome has been placed in the section of General considerations in gene editing design for CF (Lines 133-150).
Reviewer 3 Report
The review by Guoshun Wang provides a broad and above all complete overview of cystic fibrosis, genome editing techniques and the latter applied to cystic fibrosis. The review is well written and a pleasure to read. Maybe just two minor revisions, perhaps section 3 “General considerations in gene editing design for CF” would be good to put it at the end by expanding the discussion, and the presence of some figures or diagrams would further facilitate understanding and reading of the same.
Author Response
Thank you for your evaluation of our manuscript. This review is intended to update the field with the new advances of the technology and its adaptation to CF gene editing. Previous publications have provided very detailed descriptions of the gene editing mechanisms with figures and diagrams, many of which have been cited. Thus, the author decided not to provide similar illustrations.
With regarding to where to position Section 3 “General considerations in gene editing design for CF”, while putting it at the end has some merits, the author feels that keeping it at where it was serves the purpose better as a general guidance for editor design before introducing the different editors. Thus, the author respectfully requests leaving this section at the original position.